# Exploiting limited valence patchy particles to understand autocatalytic kinetics

Silvia Corezzi [1], Francesco Sciortino[2] & Cristiano De Michele [2]

Autocatalysis, i.e., the speeding up of a reaction through the very same molecule which is produced, is common in chemistry, biophysics, and material science. Rate-equation-based approaches are often used to model the time dependence of products, but the key physical mechanisms behind the reaction cannot be properly recognized. Here, we develop a patchy particle model inspired by a bicomponent reactive mixture and endowed with adjustable autocatalytic ability. Such a coarse-grained model captures all general features of an auto-catalytic aggregation process that takes place under controlled and realistic conditions, including crowded environments. Simulation reveals that a full understanding of the kinetics involves an unexpected effect that eludes the chemistry of the reaction, and which is crucially related to the presence of an activation barrier. The resulting analytical description can be exported to real systems, as confirmed by experimental data on epoxy–amine polymeriza-tions, solving a long-standing issue in their mechanistic description.

---

[1] Dipartimento di Fisica e Geologia, Universitá di Perugia, I-06123 Perugia, Italy. [2] Department of Physics, Sapienza University of Rome, I-00185 Rome, Italy. Correspondence and requests for materials should be addressed to S.C. (email: silvia.corezzi@unipg.it)

Autocatalysis is very important in life and probably played a central role in its origin on Earth[1]. The prebiotic synthesis of molecular-building blocks[2, 3], the emergence of complex behavior in networks of organic reactions[4], the establishment of self-replicating systems capable of sustaining and evolving information[5, 6] and ultimately, the transition from inanimate matter to living organisms[7], they all seem unfeasible without molecular autocatalysis. It also comes into play in biological cycles[8] and synthetically useful transformations[9, 10], as well as in the production of engineering materials, such as epoxy resins, which are formed by autocatalytic step-growth polymerization and used in many technological applications[11]. The phenomenon of autocatalysis is inherently kinetic. The fingerprint is the sigmoidal kinetic profile: the reaction is slow initially as little catalyst is present, progressively accelerates as the amount of catalyst increases and then slows down again due to reactant depletion.

The study of kinetics is key to make inferences about the molecular mechanisms of a reaction and their physical description. The comparison of reaction rates with the prediction of models is the most general method for validating a chemical description. Over the years, deviations from expected kinetics kicked off the search for new or modified mechanistic understanding of important processes, e.g., the abnormal assembly of proteins into amyloid fibrils implicated in human diseases[12] or the curing of epoxy resins whose control is important, among others, for constructing mechanically optimized structural components[13]. But also, the speed at which products are formed in a reaction impacts all other reactions that are using one of them as a reagent and represents an important controlling factor for the overall process. Despite the pervasiveness of autocatalytic phenomena in nature and technology, and despite the importance of studying the kinetics of chemical processes, autocatalytic aggregation has so far remained out of reach of realistic numerical simulations. This prevents the underlying physical mechanisms from being recognized and full understanding of the process from being achieved. Rate equations, for example, which describe models where reacting units are viewed as a gas of point-like entities, have been solved numerically for epoxy network formation[14, 15] and amyloid fibril self-assembly[16–18] but within this approach crowding (then, excluded volume) effects and the role played by shape anisotropy of the constituent particles are neglected; in addition, no consideration is given to the fact that real monomeric units react and bond directionally as a result of well-defined coordination geometries, after they have diffused in space.

Patchy particle models, which are designed learning from nature, can be thought of as an archetype of real monomers and their individual motion is an integral part of the bonding process. While these models have been predominantly used in the field of self-assembly of colloids, their range of applicability is much wider, ranging from associating and network-forming liquids[19, 20] to chemically reactive particles[21]. Their flexibility originates from the possibility of implementing a limited number of reactive bonds by geometrically enforcing the single bond per patch condition[22]. Nevertheless, so far they completely lack autocatalysis and as such cannot be used to shed light on the physics behind autocatalytic aggregation phenomena. Here we describe how to endow a patchy particle model with the ability to aggregate autocatalytically and we show for this model the emergence of an unexpected kinetic mechanism, that eludes the chemistry of the reaction. To this end we focus on a binary mixture of mutually reactive monomers represented as hard homogeneous ellipsoids, surface-decorated with a small number of bonding sites arranged in a fixed geometry and interacting via a squared-well attractive potential (Fig. 1). With a view to comparing the simulation results with experiments of formation of

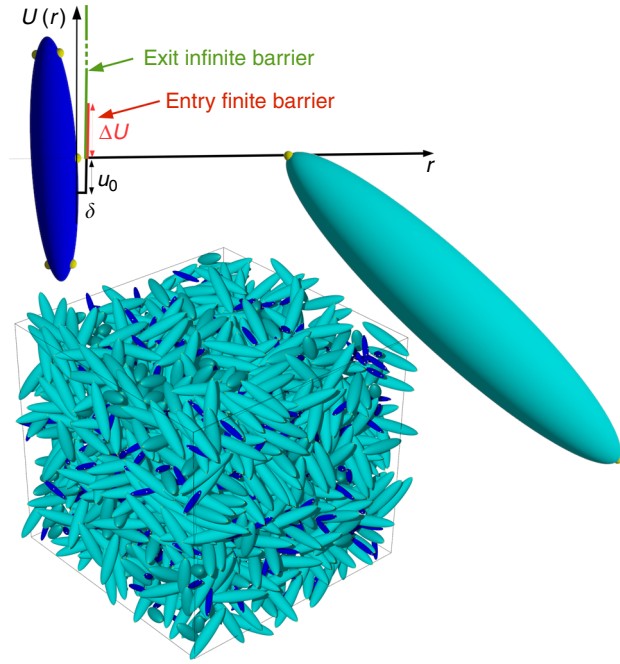

**Fig. 1** The model. Graphic description of the two types of hard ellipsoids composing the simulated system, and snapshot of the system in the initially monomeric state. The centers of the small (yellow) spheres locate the bonding sites on the surface of the hard-core particle. The site–site interaction, sketched as a function of the inter-site distance $r$, is modeled as an attractive square-well potential complemented by a repulsive entry barrier of finite height (in red) and an infinite repulsive exit barrier (in green). Bonds can be formed only between small (blue) and big (cyan) particles

effectively irreversible strong covalent bonds between epoxy and amino reagents, once a bond is formed, bond breaking is prevented by an infinite barrier which constrains the reacted sites to remain trapped in the well. The model offers the possibility to include and modulate the height of an energy barrier that must be overcome to enter the potential well and which, according to current reasoning, would only be expected to slowdown bond formation depending on its relative value compared to the thermal energy $k_B T$—$k_B$ being the Boltzmann's constant. On the contrary, we observe anomalous rate of bond formation dependent on the barrier and we demonstrate that this kinetic behavior has a dynamical origin connected to the motion of particles around the obstacle to be overcome. Since autocatalysis according to a mean-field treatment can be traced back to a continuous reduction of energy barrier as the reaction progresses, this effect which is observed during non-autocatalytic aggregation of the ellipsoidal particles, re-emerges in the kinetics of the model in which particles are endowed with the ability to bond autocatalytically, and guides to an analytical description of the observed kinetics which is exportable to real systems, as confirmed by experimental data on epoxy–amine polymerizations. Our study thus provides a powerful tool to investigate under realistic conditions autocatalytic aggregation processes, including those occurring in crowded environments.

## Results

**Autocatalysis scenarios**. Catalysis is the increase in the rate of a chemical reaction due to the participation of an additional substance, called catalyst, not consumed in the reaction. According to transition state theory[23], in order to transform into products the

reactants must form a temporary intermediate with a higher free energy: a catalyst works by providing an alternative reaction pathway involving a different transition state with a lower activation free energy. The rate of reaction increases because more reactant molecules collide with enough energy to surmount the smaller energy barrier. Autocatalysis is the catalysis provided by a reaction product. In a generic autocatalytic scheme, compound $C$ facilitates a chemical reaction between compounds $A$ and $B$, in which a second $C$ molecule is formed. We rationalize this fact as that every time a new bond is formed between a pair of mutually reactive sites of type $A$ and $B$, this creates in the system a catalyst unit $C$, existing as a separate species or as a side-group attached to one reacted molecule, whose proximity in turn is able to lower the free energy barrier between another pair of unreacted sites. The reaction between $A$ and $B$ species might naturally start via a slower noncatalyzed pathway, or might be too slow to occur naturally and be initiated by a small amount of catalyst initially present as impurities or intentionally added. These are the basic autocatalytic scenarios of practical importance to which we refer, and which we aim to reproduce within a simple patchy particle model where bonds irreversibly form between complementary reactive sites, giving rise to larger and larger particle aggregates in the form of branched clusters or networks.

It has been previously noticed[24–27] that owing to the random nature of the growth mechanism in step-wise aggregation, the system's evolution slaved to the bonding process is expected to proceed along a sequence of equilibrium states only controlled by the number of formed bonds, so that when the fraction of bonds is $p$, the number $n(s)$ of clusters made of $s$ particles can be calculated following statistical mechanics prescriptions[24], independent of the time evolution of $p$. The presence of autocatalytic mechanisms of aggregation, on the other hand, expresses itself in the evolution of $p$ over time. Therefore, to find out the rule for a proper design of autocatalytic patchy particles, first we develop a statistical treatment for the rate equation of an autocatalytic, irreversible aggregation process.

**Autocatalytic chemical rate constant**. Within a mean-field approach, the rate of an irreversible bimolecular reaction is expected to be proportional to the product of concentration of the unreacted sites, i.e., $dp/dt = k_{overall}(1-p)^2$, where $p$ is the fraction of bonds formed at time $t$, and $k_{overall}$ is a proportionality coefficient that incorporates all information on the aggregation process, by taking into account both the chemistry of the bonding act and the diffusive properties of the molecules to which the reactive sites are attached. Since $1/k_{overall} = \tau_{overall}$ represents the characteristic time needed to form a single bond, it is reasonable to assume it is given by the average time needed for two molecules to diffuse through the sample and encounter, plus the time needed for the bonding act to take place, i.e., $\tau_{overall} = \tau_{diff} + \tau_{chem}$, where $\tau_{diff} = 1/k_{diff}$ with $k_{diff}$ a diffusive rate constant related to the diffusion timescale of the system, and $\tau_{chem} = 1/k_{chem}$ with $k_{chem}$ a chemical rate constant dependent on the intrinsic reactivity of the sites. By simply considering the bonding act that would occur without diffusional restrictions, we note that the overall number of bonds per unit time is the sum of those formed in close proximity to and far from a catalytic site, and therefore we write the chemical rate constant as the sum of an autocatalytic ($k_{auto}$), a catalytic ($k_{cat}$), and a noncatalytic ($k_{non}$) contribution, i.e., $k_{chem} = P_a k_{auto} + P_c k_{cat} + [1 - P_a - P_c]k_{non}$, where $P_a$ and $P_c$ are the probabilities that the bonding act occurs via the autocatalytic and the catalytic pathway of reaction, respectively. These probabilities correspond to the probability of finding the autocatalyst and the initial catalyst within an active distance from the reactive sites. Assuming for both catalyst types

that their spatial distribution is uniform, and that the catalyst unit has to be very close to the reacting sites in order to catalyze their reaction, we calculate these probabilities (Supplementary Note 1) and find that the chemical rate constant, in both the autocatalytic scenarios mentioned above, can be given as a linearly increasing function of the extent of reaction, i.e., $k_{chem} = k_1 + k_2 p = k_1(1 + \xi p)$, by setting $\xi = k_2/k_1$. Specifically, for reactions that naturally start in the absence of any added catalyst, $k_1$ corresponds to the noncatalytic rate constant and $k_2$ is proportional to the difference between the autocatalytic and the noncatalytic one; in contrast, for reactions initiated by an added catalyst, $k_1$ and $k_2$ are respectively proportional to the catalytic and autocatalytic rate constant. In both cases, the parameter $\xi$ is related to the reduction of energy barrier for single bond due to the action of the autocatalytic agent (see Supplementary Note 1)—we herein call $\xi$ the "autocatalytic strength" of the reaction.

By introducing an effective free energy barrier having the form $\Delta G = \Delta G_0 - \beta^{-1}\ln(1 + \xi p)$ with $\beta = 1/k_B T$, and an effective time constant $\tau_0$, the chemical rate constant is written $k_{chem} = \tau_0^{-1}e^{-\beta\Delta G}$, that is the same as when the activation barrier for bonding any pair of reactive sites is $\Delta G$. Autocatalysis is responsible for the progressive reduction of such a barrier, on increasing the number of bonds, from an appropriate initial value $\Delta G_0$ which represents the activation barrier for bond formation not mediated by the autocatalyst, in the case of reactions that start naturally, and represents the activation barrier for bond formation mediated by an added catalyst, in the case of reactions otherwise blocked (see Supplementary Note 1).

**Implementation of autocatalysis**. On the basis of the previous treatment, the mechanism of autocatalysis can be effectively traced back to a progressive reduction of the free energy barrier for bond formation as the reaction proceeds. According to this prescription, we modify the patchy particle model previously introduced[21] to make the particles able to aggregate autocatalytically. To facilitate comparison with available experiments the particles are chosen to resemble a stoichiometric mixture of (bifunctional) epoxy and (pentafunctional) amino reagents, and the characteristic features of the interaction potential to qualitatively mimic covalent-bond formation between mutually reactive monomers (Methods section). Since in the model the entropy barrier for bonding two reactive sites only depends on geometrical factors, the variation of $\Delta G = \Delta U - T\Delta S$ as the reaction progresses is charged on $\Delta U$. We therefore implement the autocatalytic mechanism by introducing in the inter-site potential an incoming energy barrier (Fig. 1) which depends on the number of bonds present in a system's configuration and which progressively reduces on increasing $p$, starting from the initial value $\Delta U_0$, according to

$$\Delta U = \Delta U_0 - \beta^{-1}\ln(1 + \xi p) \qquad (1)$$

The parameter $\xi$, i.e., the autocatalytic strength, quantifies the reduction rate. Note that, according to the Boltzmann distribution, the probability to overcome the barrier becomes $\sim e^{-\beta\Delta U_0}(1 + \xi p)$, i.e., grows linearly as the reaction proceeds, thus establishing a positive feedback mechanism. Such a system combines the representativity of basic autocatalytic processes which are of real interest, with the advantage of a mean-field approach. It thus provides the simplest treatable model for the investigation of autocatalytic aggregation under realistic conditions.

We study the kinetics of the model for several values of $\beta\Delta U_0$ and different values of $\xi$. The resulting curves of $p$ versus time for $\beta\Delta U_0 = 4$ are shown in Fig. 2. For all $\xi$, the rate of bond

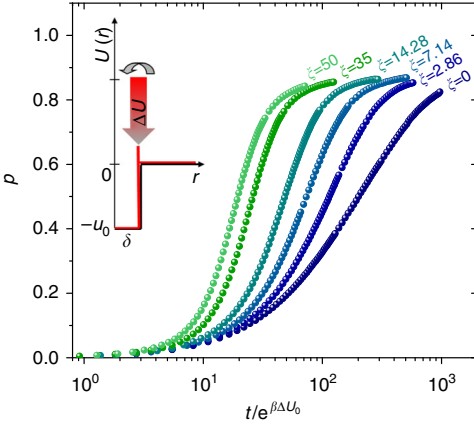

**Fig. 2** Kinetics of autocatalytic reactions. Kinetic curves, $p$ versus time, at fixed temperature and energy barrier ($\beta\Delta U_0 = 4$) for different values of autocatalytic strength $\xi$, as indicated. Time is scaled by $e^{\beta\Delta U_0}$ to highlight speed-up of the reaction due to autocatalysis. The profile of the site–site potential for bond formation, including its qualitative behavior during reaction, is sketched in the inset as a function of the inter-site distance $r$ (Methods section)

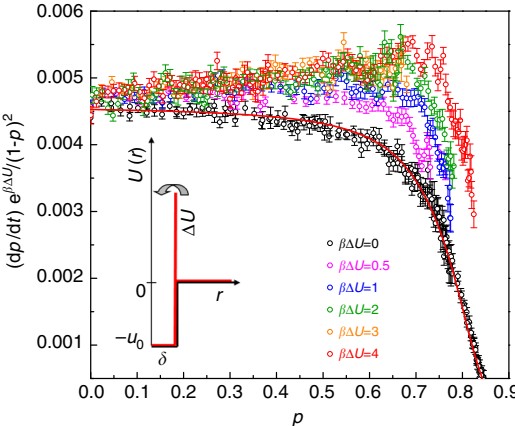

**Fig. 3** Barrier effect. Reaction rate $dp/dt$ scaled by $(1-p)^2 e^{-\beta\Delta U}$, for different values of energy barrier, as indicated. For each $\beta\Delta U$, symbols are mean ± SEM of the results of 40 independent simulation runs. The solid line is the description of the data in the case without barrier, using $dp/dt = k_{overall}(1-p)^2$, with $k_{overall} = (1/k_c + 1/k_d D)^{-1}$, where $D$ is the $p$-dependent average diffusion coefficient of a particle in the system (see next Fig. 4a)

formation passes through a maximum at an intermediate reaction time $t > 0$, giving the kinetic curve the characteristic profile observed in many real systems.

**Barrier effect.** To understand these profiles and link the kinetics with the physics of the aggregation process, we perform additional simulations with fixed-barrier particles, at several values of $\beta\Delta U$. An excellent description (solid line in Fig. 3) of the overall kinetics in the case $\beta\Delta U = 0$, already studied in ref. [28], is obtained according to the mean-field description mentioned above, by assuming a diffusive rate constant directly proportional to the $p$-dependent average diffusion coefficient, $D$, of a particle in the system ($k_{diff} = k_d D$). In this case, two sites bond each other when achieving the bonding distance $\delta$, whatever energy they have. In contrast, for $\beta\Delta U > 0$ two colliding sites not always have enough energy to overcome the barrier, and an expected number $\mathcal{N} = e^{\beta\Delta U}$ of collisions are needed before a successful event occurs (Methods section). Once two sites have undergone an unfruitful collision the particles to which they belong go back to diffuse through the sample and the next encounter, producing another attempt to overcome the barrier, will take on average as much time as the previous one. The average time to form a bond, $\tau_{overall} = \tau_{chem} + \tau_{diff}$, gets longer by such a number of trials $\mathcal{N}$. As a result, the chemical rate constant of the system, which is the reciprocal of $\tau_{chem}$, is expected to reduce with the height of barrier as $k_{chem} = k_c/\mathcal{N} = k_c e^{-\beta\Delta U}$, with $k_c$ the value without barrier, and analogously the diffusive rate constant, which is the reciprocal of $\tau_{diff}$, is expected to reduce as $k_{diff} = k_d D/\mathcal{N} = k_d D e^{-\beta\Delta U}$, with $k_d$ the same as without barrier provided that the average diffusion coefficient does not depend on the barrier itself. Therefore, it is $k_{overall} = (1/k_{chem} + 1/k_{diff})^{-1} = (1/k_c + 1/k_d D)^{-1} e^{-\beta\Delta U}$. With these expectations, dividing the reaction rate by $(1-p)^2 e^{-\beta\Delta U}$ should produce a master curve of the data as a function of $p$, providing a direct visualization of the coefficient $(1/k_c + 1/k_d D)^{-1}$. This representation is shown in Fig. 3, in which two kinds of deviations from a master curve are observed, respectively at lower and higher $p$ values. In the early stages of reaction, when the diffusivity is high and the system is in chemically controlled regime, $k_{overall}$ is approximated by $k_{chem}$ and the data should flatten out, towards the value $k_c$. For $\beta\Delta U > 0$, however, this behavior is not observed but the slope of the data increases on increasing the barrier, up to a saturation value. This

effect cannot be ascribed to diffusional limitations, because these latter intervene at a later stage and contribute to decrease, not to increase the reaction rate. Thus, a "barrier effect" emerges in the chemically controlled kinetics: a barrier to overcome produces a rate of bond formation which does not simply scale exponentially with the height of the barrier, giving rise to an enhancement over the expected value. Since we can empirically compensate for the rate excess observed in this range by dividing the rate by $(1-p)^n e^{-\beta\Delta U}$ rather than by $(1-p)^2 e^{-\beta\Delta U}$, with $n < 2$ an exponent that depends on $\beta\Delta U$, this barrier-induced effect is accounted for analytically by introducing a corrective factor to $k_{chem}$, dependent on the barrier and the extent of reaction, given by $(1-p)^{n-2}$.

At a later stage, when the encounter probability between particles has dropped significantly and the system is in diffusion-controlled regime, $k_{overall}$ is approximated by $k_{diff}$ and the data in Fig. 3 should collapse on the $p$-dependent quantity $k_d D$ evaluated for the case without barrier. The curves for $\beta\Delta U > 0$, however, increasingly depart from the curve with $\beta\Delta U = 0$ and the departure starts at increasingly high values of $p$, indicating that the diffusion-controlled regime dominates at a higher extent of reaction the higher is the barrier. Thus, a "barrier effect" also emerges in the diffusion-controlled kinetics: a barrier to overcome is associated to an enhanced diffusive rate constant compared to the value expected by scaling exponentially with the height of the barrier. To investigate the origin of such enhancement we analyze the average diffusion coefficient $D$, calculated from the long-time limit of the mean-squared displacement averaged over all the individual particles composing the system (see Methods section and Supplementary Fig. 1). The comparison between $\beta\Delta U = 0$ and 4 is shown in Fig. 4a, and no difference in the behavior of $D$ versus $p$ is surprisingly found. Consistent with this fact, we also find that the distribution $n(s)$ of the size $s$ of particle clusters in the system is not altered by the presence of activation barriers, not when the height of barrier remains fixed (Fig. 4b), not when it progressively reduces (Fig. 4c), suggesting a dynamical origin for the observed barrier-induced effect that does not involve changes in the overall structure. This is confirmed by the static structure factor $S(q)$, where $q$ is the modulus of the wave vector $\mathbf{q}$ (Supplementary Fig. 2). Based on these results, the enhancement of $k_{diff}$ cannot be ascribed to variations of $D$, but rather implies a modification of $k_d$, dependent

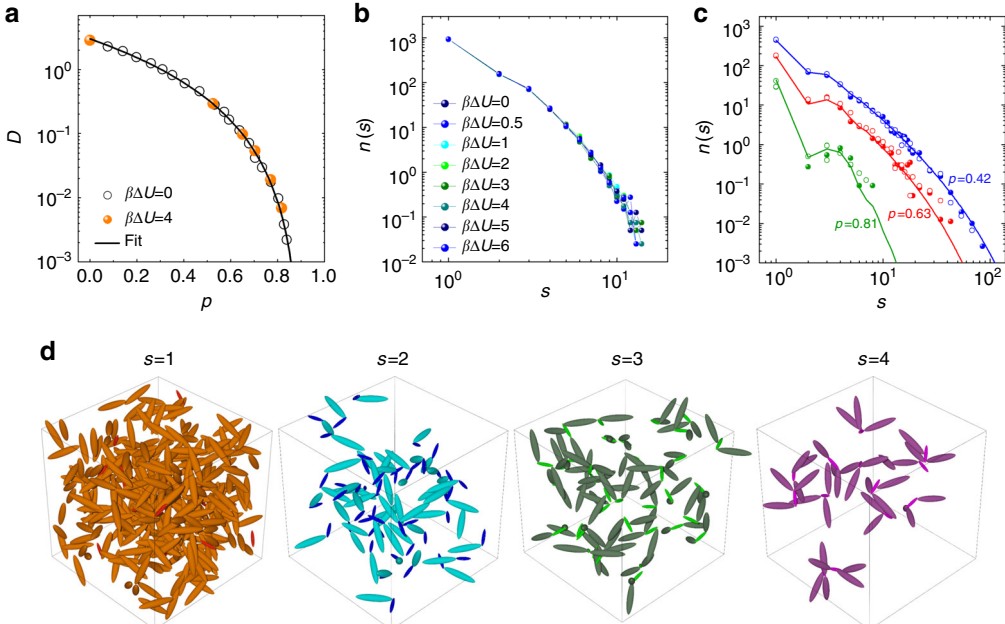

**Fig. 4** System's diffusivity and cluster size distribution. **a** Average diffusion coefficient $D$ determined from the particle mean-squared displacements (Methods section) for the system without barrier and with barrier. The $p$ dependence is described by $D = D_0[(p_0 - p)/p_0]^\gamma$ with $\gamma = 2.69$, $p_0 = 0.902$, and $D_0$ the diffusion in the initially unbonded state. **b** Distribution $n(s)$ of the size $s$ of clusters present at $p = 0.2$ in systems with fixed barrier (as indicated), providing evidence that the cluster size distribution is independent of both presence and height of the barrier. **c** Cluster size distribution during non-autocatalytic reaction with barrier $\beta\Delta U = 4$ (closed symbols) and during autocatalytic reaction of equal initial barrier (open symbols). Solid lines represent Flory-Stockmayer predictions[24]. The comparison is shown at three values of $p$, before and after percolation ($p_{gel} = 0.5$). No difference emerges, providing evidence that the distribution of clusters at any given $p$ is independent from the autocatalytic nature of the reaction. **d** Populations of clusters of different size, from monomers to tetramers, in the system reacted up to $p_{gel} = 0.5$. In each population, big and small particles are distinguished by color

on the barrier. It is important to note that the reaction rate in the high $p$ region is not scalable with the corrective factor found at lower $p$ values (Supplementary Fig. 3), revealing that the same activation barrier is causing a different enhancement of $k_{chem}$ and $k_{diff}$ in the system. For $k_{diff}$ we assume a corrective factor $k_0$, dependent on $\beta\Delta U$. To quantify these observations, we reproduce the reaction rate over the entire range of $p$ for all $\beta\Delta U$ by using

$$k_{chem} = k_c(1-p)^{n-2}e^{-\beta\Delta U}$$
$$k_{diff} = k_d k_0 D e^{-\beta\Delta U} \qquad (2)$$

in which $k_c$ and $k_d$ are obtained from the case without barrier, $D$ versus $p$ is known from the mean-squared displacement, and the fit only adjusts the exponent $n$ and the constant prefactor $k_0$ for each curve.

Figure 5a compares the results of the best-fit for $\beta\Delta U = 0$ and 4, showing that in the second case the description requires a value of $n$ much lower than 2 and a value of $k_0$ much higher than 1. The $\beta\Delta U$ dependence of these parameters is shown in Fig. 5b, c: $n$ exponentially decays with the height of barrier as $e^{-\beta\Delta U}$, towards a limiting value which is rapidly reached at $\beta\Delta U \approx 2$; $k_0$, in the range of barrier explored, exponentially increases as $e^{b\beta\Delta U}$ with $b$ significantly lower than 1.

In its apparent simplicity, the simulation unveils a novel kinetic effect, related to the presence of activation barriers. So far this effect has been ignored, despite the fact that it may apply to many situations. To go deeper into the physics of this effect, we consider a model, based on the solution of Smoluchowski equation of diffusion in three dimensions[29], which aims to describe in a simplified way the growth process of a sink (due to clustering) in the presence of a barrier for particle

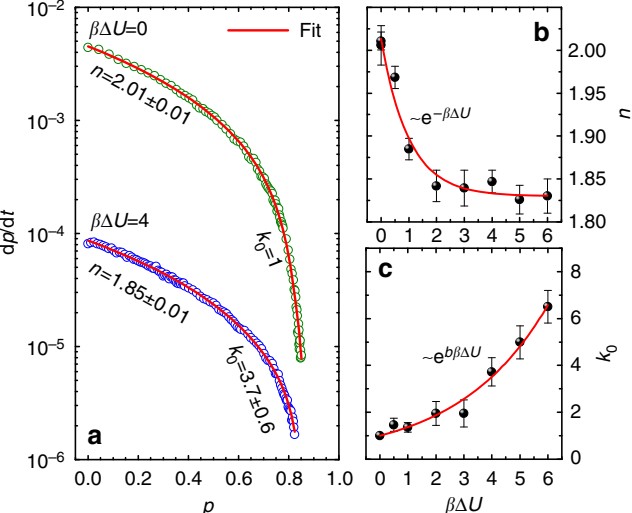

**Fig. 5** Dependence of non-autocatalytic kinetics on the height of barrier. **a** Comparison between the kinetics of the model with barrier and without barrier. The data are fitted by $dp/dt = k_{overall}(1-p)^2$, where $k_{overall} = (1/k_{chem} + 1/k_{diff})^{-1}$ with $k_{chem}$ and $k_{diff}$ given by Eq. 2, by weighting each datapoint with its standard error (not shown since within the symbol size). By using $k_c = 4.56 \cdot 10^{-3}$ and $k_d = 1.015$ obtained with $\beta\Delta U = 0$, and $D(p)$ obtained from the mean-squared displacement, the fit only adjusts two parameters, $n$ and $k_0$, for each $\beta\Delta U > 0$. The uncertainty of the best-fit parameters is reported in **b** and **c** with error bars. **b** The barrier dependence of the exponent $n$ is analytically described by $n = (n_0 - n_\infty)e^{-\beta\Delta U} + n_\infty$, with $n_0 = 2.01 \pm 0.01$ and $n_\infty = 1.830 \pm 0.006$. **c** The barrier dependence of $k_0$, over the investigated range, is approximated by $k_0 = e^{b\beta\Delta U}$, with $b = 0.31 \pm 0.02$

adsorption (see Supplementary Note 2). In this model, although the overcoming of the barrier is an activated process, the adsorption rate of the diffusers does not scale as $e^{-\beta \Delta U}$, but it appears a corrective factor always greater than 1, dependent parametrically on both the barrier height and the extent of reaction (Supplementary Fig. 4c), in qualitative agreement with what is observed in the simulation model. Since the system in the chemically and in the diffusion-controlled regime is in very different microscopic conditions, it is reasonably expected that different rate-corrective factors have to be used to model adequately the two regimes. In particular, Supplementary Fig. 4c qualitatively suggests that in the early and intermediate stages of reaction, when clusters are experiencing rapid growth, the barrier dependence of the corrective factor tends to saturate and shows a more pronounced $p$ dependence. In the late stages of reaction, when new bonds mainly incorporate small clusters into the percolating gel network which can be assimilated to a negligibly growing sink, the $p$ dependence is expected to vanish and the barrier dependence dominates the corrective factor.

**Autocatalytic kinetics**. We now reconstruct the autocatalytic kinetics for all the reactions simulated, building on the kinetics of the non-autocatalytic model with barrier, by replacing into Eq. 2

the fixed value of barrier with a barrier that progressively reduces on increasing $p$ according to Eq. 1. With this replacement, one obtains

$$
\begin{aligned}
k_{\text{chem}} &= k_c (1-p)^{n-2}(1+\xi p)e^{-\beta \Delta U_0} \\
k_{\text{diff}} &= k_d k_0 D(1+\xi p)e^{-\beta \Delta U_0}
\end{aligned}
\tag{3}
$$

where $\Delta U_0$ is the initial value of barrier. Note that because of the change of the barrier with $p$, the exponent $n$ and the prefactor $k_0$ also change during the reaction as

$$
\begin{aligned}
n &= n_\infty + (n_0 - n_\infty)(1+\xi p)e^{-\beta \Delta U_0} \\
k_0 &= (1+\xi p)^{-b}e^{b\beta \Delta U_0}
\end{aligned}
\tag{4}
$$

thus incorporating in the autocatalytic rate equation the barrier effect previously discussed, and producing non-negligible corrections to the kinetic profile over the entire $p$ range. In Fig. 6 the solid lines are the kinetic curves calculated for different values of $\beta \Delta U_0$ and $\xi$ using only parameters borrowed from non-autocatalytic simulations. Although no fit parameters are used, in all cases the agreement with the numerical simulation is excellent, demonstrating that the autocatalytic kinetics is determined by how the system behaves in non-autocatalytic conditions, combined with the way the barrier effectively decreases.

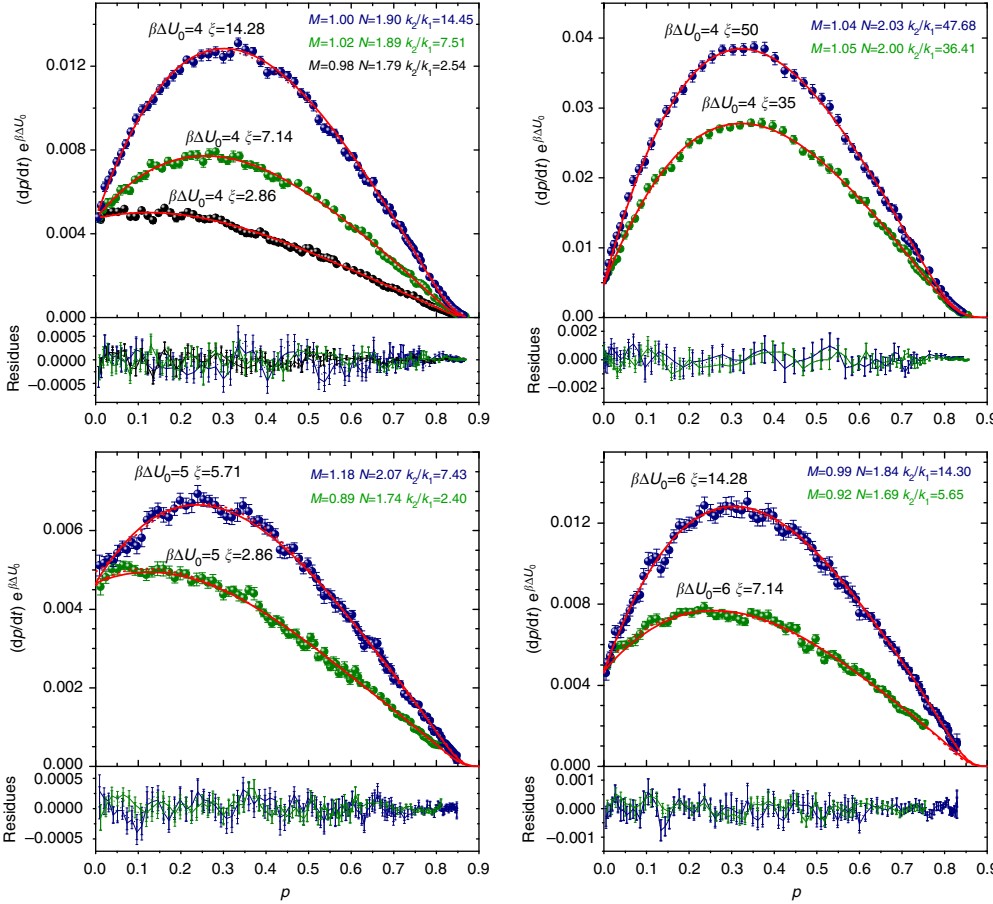

**Fig. 6** Description of the autocatalytic kinetics. The $p$ dependence of the rate $dp/dt$ (scaled by $e^{-\beta \Delta U_0}$ to facilitate comparison) at different autocatalytic strengths $\xi$ and initial energy barriers $\beta \Delta U_0$. The solid lines are the predicted kinetics, i.e., $dp/dt = k_{\text{overall}}(1-p)^2$ where $k_{\text{overall}} = (1/k_{\text{chem}} + 1/k_{\text{diff}})^{-1}$, with $k_{\text{chem}}$ and $k_{\text{diff}}$ given by Eqs. 2 and 4, and using $D(p)$ obtained from the mean-squared displacement. All the parameters are known from non-autocatalytic reactions. The dotted lines (in some cases they cannot be distinguished from the solid lines) represent the best-fit of the simulation data with $k_{\text{chem}}$ replaced by the Kamal's rate constant, i.e., $k_{\text{chem}} = (k_1 + k_2 p^M)(1-p)^{N-2}$ with $k_1$, $k_2$, $M$, and $N$ free-parameters independent of $p$. The values of $M$, $N$, and $k_2/k_1$ associated to each dotted line is indicated. Below each frame, in the same color as the symbols, the residues with respect to the predicted kinetics are shown as solid lines, and with respect to the best-fitted Kamal equation as dotted lines. Different panels refer to different values of $\beta \Delta U_0$

This analysis explains, on a physical basis, the equation that describes the kinetics of bond formation in the model of autocatalytic patchy particles; yet this analysis is not offering, per se, substantial evidence that the model is realistic and that the effect of barrier may come to play a visible role in real systems. As a next step, we provide evidence that the model equation derived through the simulation study closely reproduces a number of experimental data, in which the barrier effect shows up clearly.

**Comparison with experiments**. To provide evidence that the simulation model is able to reproduce the kinetics of bond formation in real situations, we consider the process of epoxy–amine cure, very important in material science. Depending on the reactants and the curing conditions, it is not unusual to be in the case where the addition of an amino hydrogen to the epoxy ring represents the exclusive reaction and all amino hydrogens have equal degree of reactivity toward epoxy groups. The addition reaction results in the formation of a hydroxyl group that may become involved in a hydrogen-bonded transition state in which an unreacted epoxy ring is more sensitive to nucleophilic attack, thus considerably promoting further interaction with amines[30–32]. The reaction between epoxy and amino species, however, is too slow to occur naturally, and it is initiated by any hydrogen-bond donor molecule (XH) initially present in the form of moisture or impurities. The XH molecules, together with the OH groups in the reaction products, are not consumed in any side-reactions and act as true catalysts and autocatalysts, respectively. Therefore, the reaction is an example of autocatalytic reaction initiated by an added catalyst. Figure 7 shows a qualitative comparison of experimental data for selected epoxy resins reacted at various temperatures with data from the simulation, providing a good anticipation of the ability of the model to capture the essentials of the kinetics of these systems.

For a stringent test, we go a step further and quantitatively reproduce the experimental data of six epoxy resins with the model equation derived through the simulation study. To this purpose, the expression of $k_{diff}$ requires prior adjustment to account for the different $p$ dependence of $D$ in real systems with respect to the numerical case, and for the lack of direct access to

D. To this end, the expression of $k_{diff}$ in the modeling of epoxy resins is replaced by $k_{diff} = k_d \tau^{-\lambda}$, where $\tau(p)$ is the structural relaxation time, for example measured by dielectric[33, 34] or photon-correlation spectroscopy[35, 36], and $0 < \lambda < 1$ is a system-dependent exponent. Both $k_d$ and $\lambda$ are independent of $p$ and reaction $T$. This avoids the introduction of redundant fit parameters; it also takes into account the temperature dependence of $k_{diff}$ through the $T$ dependence of $\tau$ and, at the same time, it assumes $D$ is related to $\tau$ via a fractional power–law relationship, as observed in many fluids close to vitrification[37, 38]. As for $k_{chem}$, we use the analytical expression in Eq. 3, with $n$ given by Eq. 4, by recalling that $\xi = k_2/k_1$. Since $k_1$ and $k_2$ are directly proportional in the present scenario to the rate constants associated respectively to the catalytic and autocatalytic pathways of reaction (Supplementary Note 1), they can also be written as $k_1 = A e^{-\beta \Delta U_c}$ and $k_2 = B e^{-\beta \Delta U_a}$, with $\Delta U_c$ and $\Delta U_a$ the enthalpy barriers for bond formation mediated by the initially present catalyst (XH) and by the autoproduced catalyst (OH), being the entropy barriers incorporated into the prefactors $A$ and $B$. Using $n_0 = 2$ (the value of $n$ at $\Delta U = 0$) as found in simulation, we describe the kinetics of each epoxy system at several reaction temperatures with five parameters for the chemically controlled rate, i.e., $\Delta U_c$, $\Delta U_a$, $A$, $B$, $n_\infty$, and two parameters for the diffusion-controlled rate, i.e., $k_d$ and $\lambda$, all independent of reaction $T$ and, therefore, shared by all isotherms. It is remarkable that with a single set of parameters we are able to account not only for the $p$ dependence of the reaction rate but also for its temperature dependence. Figure 8 shows the excellent agreement, for each system, of the experimental data with the result of a simultaneous fit procedure. Table 1 summarizes these results. Similar values of enthalpy barrier are obtained independently of the amine used in the cure process; in all cases, moreover, the value of $\Delta U_c$ is equal or very close to $\Delta U_a$, supporting the idea that the initially present catalyst XH is mostly formed by OH groups, likely due to moisture. More interestingly, the value $n_\infty < 2$ which is crucial to obtaining a successful description, stands out as a manifestation of the barrier effect.

These findings resolve a long-standing issue concerning the kinetics of these materials. Experimental data systematically deviate from the kinetics that would be expected, so that a phenomenological rate equation with $k_{chem} = (k_1 + k_2 p^M)(1 - p)^{N-2}$, which is known as Kamal equation[39], has been used as a replacement. This equation is extremely flexible, provided that $M$ and $N$ are left free to assume fractional and temperature dependent values; but these exponents still remain without physical explanation. It is widely accepted that a less simplified reaction scheme is hiding behind values of these exponents respectively different from 1 and 2, yet no study has been able to substantiate this guess[40–42]. Our results indicate that the observed kinetics can be fully explained on a physical basis without any need to complicate the reaction scheme, by means of what we called "barrier effect". Moreover, there is no guarantee that the empirical Kamal's modification preserves $k_1$ and $k_2$ in their original meaning of catalytic and autocatalytic rate constants and, therefore, no guarantee that their Arrhenius plot is providing correct estimates of the associated activation energy. Figure 8 shows that the fit strategy using the Kamal equation indeed provides a comparably good description of each curve; however, it derives activation energies differing up to 50% from those obtained with our model equation. To further demonstrate this point, we fit the simulation data in Fig. 6 by using the Kamal equation, according to the common practice in the experimental literature, and find an erroneous determination of $k_2/k_1 = \xi$ in several cases, over or under-estimating the true value of $\xi$ up to 30%, despite an equally good description of the data as compared to our equation.

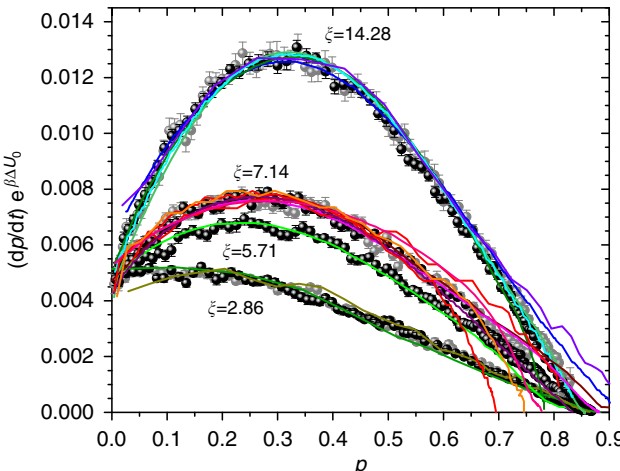

**Fig. 7** Qualitative comparison with experimental data. Symbols: reaction rate scaled by $e^{-\beta \Delta U_0}$ obtained from simulation Colored lines: experimental data for stoichiometric formulations of epoxy–amine resins, scaled by an arbitrary factor. Discrepancy in the high $p$ region depends on the different $p$ dependence of the diffusion coefficient in experimental reality with respect to the simulation model

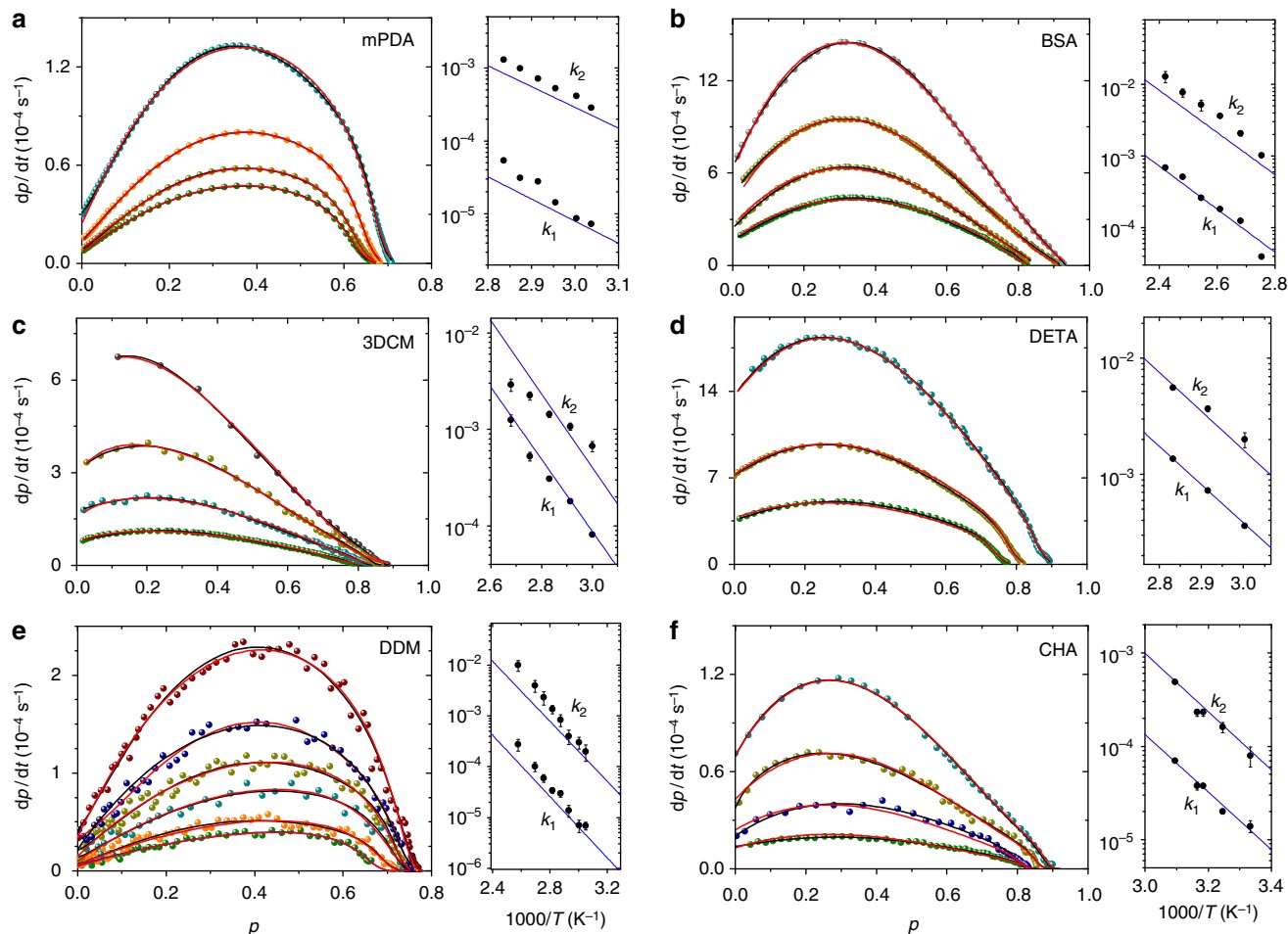

**Fig. 8** Quantitative comparison with experimental data. Colored symbols are experimental data—taken from **a** ref. [34], **b** ref. [51], **c–f** ref. [52]— for the kinetics of diglycidyl ether of bisphenol-A (DGEBA) reacted at different temperatures with different amines (1,3-phenylenediamine ($m$PDA), 4-4′(1,3-phenylene-diisopropylidene) bisaniline (BSA), 4,4′-diamino-3,3′-dimethyldicyclohexylmethane (3DCM), diethylenetriamine (DETA), cyclohexylamine (CHA), 4,4′-diaminodiphenylmethane (DDM)). Each curve corresponds to a different $T$ of reaction. The red lines are obtained by fitting the data at all temperatures simultaneously using $dp/dt = (1/k_{chem} + 1/k_{diff})^{-1}(1-p)^2$, where $k_{chem} = \left(Ae^{-\beta \Delta U_c} + Be^{-\beta \Delta U_a}p\right)(1-p)^{n-2}$ with $n = n_\infty + (n_0 - n_\infty)\left(e^{-\beta \Delta U_c} + (B/A)e^{-\beta \Delta U_a}p\right)$, and $k_{diff} = k_d \tau^{-\lambda}$ with $\tau$ the $p$-dependent structural relaxation time, experimentally measured at each $T$. Assuming $n_0 = 2$, the simultaneous fit procedure over different isotherms adjusts for each system seven parameters, i.e., $\Delta U_c$, $\Delta U_a$, $n_\infty$, $A$, $B$, $k_d$, and $\lambda$. The black lines are obtained by fitting the same data at each $T$ separately, with $k_{chem}$ replaced by the Kamal's rate constant, and $k_{diff} = k_d \tau^{-\lambda}$ as in our strategy, and thus using $4\mathbb{N} + 2$ parameters, with $\mathbb{N}$ the number of isotherms. The behavior of $k_1 = Ae^{-\beta \Delta U_c}$ and $k_2 = Be^{-\beta \Delta U_a}$ corresponding to the two fit strategies is reported in the Arrhenius plot, respectively, with blue lines and black symbols

| Table 1 Fit parameters | | | |
|---|---|---|---|
| **System** | **$\Delta U_c/10^{-20}$(J)** | **$\Delta U_a/10^{-20}$(J)** | **$n_\infty$** |
| DGEBA-$m$PDA | 9.6 ± 0.1 | 9.1 ± 0.1 | 1.34 ± 0.02 |
| DGEBA-BSA | 9.6 ± 0.2 | 9.4 ± 0.1 | 1.45 ± 0.07 |
| DGEBA-3DCM | 11.9 ± 0.3 | 12.1 ± 0.3 | 1.47 ± 0.02 |
| DGEBA-DETA | 10.7 ± 0.1 | 10.9 ± 0.1 | 1.38 ± 0.02 |
| DGEBA-CHA | 9.7 ± 0.2 | 9.9 ± 0.2 | 1.65 ± 0.04 |
| DGEBA-DDM | 9.6 ± 0.2 | 9.1 ± 0.2 | 0.71 ± 0.02 |

Activation enthalpy per single bond mediated by the initially present catalyst ($\Delta U_c$) and by the autoproduced catalyst ($\Delta U_a$), and the limiting value of $n$ for different epoxy–amine systems.

## Discussion

A mixture of ellipsoidal patchy particles is a simple but effective coarse-grained model for aggregation, able to combine finite valence, specific directional bonding, selectivity of interactions and asymmetry in shape, which are the key features of many old and new-generation building blocks of self-assembling materials.

Our model is realistically inspired by the step-wise polymerization that occurs at the molecular level, but can also be considered representative of associating polymers, functionalized molecules and, moving up in the length scale of the linkable monomers, of systems with bioselective interactions and patchy colloids. Furthermore, it offers the possibility to implement equally well a reversible (physical) aggregation process, through the removal of the exit infinite barrier and the modulation of the bonding energy of reactive sites, $u_0$: once a bond is formed there is a probability of bond breaking which depends on the relative value of the thermal energy compared to the energy barrier that the reacted sites must overcome to escape the potential well[26, 27]. The study presented in this paper provides the rule to make the same model be also a paradigmatic model of autocatalytic aggregation, in which excluded volume and barrier effects are able to emerge. Indeed, understanding the autocatalytic aggregation of these particles allows us to reveal a surprising enhancement in the rate of bond formation when step-wise aggregation proceeds in the presence of activation barriers. While the Boltzmann distribution predicts that slowdown of the process grows exponentially with the ratio

of barrier to thermal energy, we observe a rate of aggregation appreciably higher than expected and dependent on the height of barrier. This increase in the rate, which corresponds to a decrease in the average time requested for two sites on distinct particles to bond, receives a contribution from reduction of $\tau_{\mathrm{diff}}$, i.e., the cumulative time requested for two particles to diffuse distances comparable to the interparticle distance as many times as the number of unsuccessful collisions with the barrier, and a different contribution from reduction of $\tau_{\mathrm{chem}}$, i.e., the cumulative time needed for neighboring particles to correctly reorient allowing two reactive sites on their surface to collide as many times as requested to bond. By using a minimal analytical model we show, in a qualitative way, that both the energy barrier in the interaction potential and the growth of clusters bearing reactive sites are crucial for this counter-intuitive kinetic effect, never emerged from existing simulation methods, but clearly detectable in real situations. Experimental polymerization rates of simple autocatalytic epoxy–amine systems showing appreciable deviation from ideal rates are fully explained by this physical, not chemistry-based effect that avoids resorting to phenomenological equations or ad-hoc modified schemes of reaction. These findings result in a fundamental caveat when interpreting kinetic data to derive mechanistic information, and demonstrate that physical mechanisms also have a role that must not be overlooked.

We expect that a model of autocatalytic patchy particles greatly enlarges the current simulation capabilities by opening the way for studying under controlled realistic conditions many processes in which not only the characteristics of the involved particles, e.g., shape and valence, determine the final outcome of the process, but also the kinetics of the involved reactions plays a crucial role, as it is when an autocatalytic reaction is going to feed other reactions or when several reactions compete for resources[43–45]; or, again, when the process of aggregation takes place into an evolving chemical environment which changes the interparticle interactions[46]. Furthermore, by giving a proof of principle of how changing in a controlled manner the energy barrier in the interaction potential is the key for controlling the kinetic mechanism of aggregation, our study provides a guideline to design colloids that self-assemble by following an autocatalytic kinetics. This raises the intriguing prospect of constructing new colloidal assemblies whose structure is defined by the timeline for linking different types of nanoscale monomers, thus moving forward the frontiers of supracolloidal chemistry[47–49].

## Methods

**Model**. We have studied a stoichiometric mixture of $N_A = 1200$ bifunctional A particles and $N_B = 480$ pentafunctional B particles, represented as uniaxial hard ellipsoids. A particles have semiaxes ($2\sigma, 2\sigma, 10\sigma$) and mass $3.4m$, while B particles are smaller and lighter, with semiaxes ($\sigma, \sigma, 5\sigma$) and mass $m$. The packing fraction is $\phi = 0.30$, a value calibrated on a realistic mixture of epoxy–amine molecules in their initially fluid state. $f_A = 2$ bonding sites on A particles (e.g., representative of epoxy groups) are located at the particle ends, while $f_B = 5$ bonding sites on B particles are located two at each end (e.g., representative of a primary amine group) and one on the equatorial line (e.g., representative of a secondary amine group). The site–site attraction is modeled as a square-well potential of depth $u_0$ and interaction range $\delta = 0.2\sigma$, complemented by a repulsive energy barrier $\Delta U$ at an inter-site distance $\delta$ (Fig. 1). Each A site can only interact with a B site. In the numerical code, two sites of different type form a bond when their distance becomes smaller than $\delta$; each site is engaged at most in one bond. Once a bond is formed, it is made irreversible by switching on an infinite barrier at distance $\delta$ between the sites involved, which constrains the distance between them to not exceed $\delta$. Clusters are defined as groups of bonded particles.

**Molecular dynamics simulations**. We have performed event-driven molecular dynamics simulations using the algorithm described in ref. [50]. The energy unit is $k_B T$, the mass unit is $m$ and the unit of time $\sqrt{10m\sigma^2/k_B T}$. The dynamic evolution of the system is studied starting from an initial configuration with no bonds between particles, at fixed temperature $k_B T = 1$. The value of $u_0$ is set to $0.1 k_B T$. We introduce a time step $\delta t = 0.05$ in reduced units, at which we periodically rescale the particle velocity to enforce a constant temperature; rescaling is performed at

time intervals of 0.1 (we checked that the simulation results do not change upon increasing this value up to 10.0). In all cases, the simulation box is taken to be periodic and the volume fixed. To quantify the fraction of formed bonds $p$ we divide the total number of bonds present in the system by the maximum number of possible bonds $f_A N_A$ (or equivalently by $f_B N_B$). All the simulation results presented are the average of independent runs of 40 independently equilibrated starting configurations.

The implementation of autocatalysis is performed by re-calculating at each simulation step the site–site energy barrier used for the next step of the simulation as $\Delta U = \Delta U_0 - \beta^{-1} \ln(1 + \xi p)$, with $\Delta U_0$ the value at $p = 0$ and $\xi$ the autocatalytic strength.

**Diffusion coefficient calculation**. At any given time during the simulation, when the average fraction of bonds over independent configurations is $p$, positions and velocities of all particles are copied and used to start a new simulation in which the bonding pattern is frozen by switching on an infinite barrier at distance $\delta$ in the inter-site potential in place of the finite barrier $\Delta U$. In this new simulation, the formed clusters remain free to move, while being prevented from bonding other clusters. The mean-squared displacement of all the particles in the system over the time period $t$ is computed as $\langle \Delta r^2(t) \rangle = \langle \sum_{i=1}^{N} |\mathbf{r}_i(t) - \mathbf{r}_i(0)|^2 / N \rangle$, where $\langle \cdot \rangle$ denotes an ensemble average, $\mathbf{r}_i$ is the position vector of a particle and $i$ labels the $N = N_A + N_B$ particles of the system. At the given $p$, the average diffusion coefficient $D$ is then calculated as

$$D = \lim_{t \to \infty} \langle \Delta r^2(t) \rangle / 6t \qquad (5)$$

evaluated where a possible subdiffusive behavior is over and a linear increase in time is established. This simulation protocol allows us to determine the diffusion properties of the system in a manner not affected by the ongoing aggregation process.

**Attempts to overcome the barrier**. The average time $\tau_{\mathrm{overall}}$ to form a bond is assumed to increase proportionally to the average number $\mathcal{N}$ of unsuccessful attempts to overcome the energy barrier $\Delta U$ in the inter-site potential. To estimate this number we note that the probability of an unsuccessful attempt according to the Boltzmann distribution is $1 - e^{-\beta \Delta U}$, and hence the probability of $n$ unsuccessful attempts will be $[1 - e^{-\beta \Delta U}]^n$. The average number of these attempts $\langle n \rangle \equiv \mathcal{N}$ is then calculated as

$$\mathcal{N} = \frac{\sum_{n=1}^{\infty} n [1 - e^{-\beta \Delta U}]^n}{\sum_{n=1}^{\infty} (1 - e^{-\beta \Delta U})^n} = e^{\beta \Delta U} \qquad (6)$$

**Data availability**. Data supporting the findings of this manuscript are available from the corresponding author upon reasonable request.

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

## Acknowledgements

S.C. acknowledges support from the project Scientific Data & Computing "CarESS" (Università di Perugia, D.R. n. 597). C.D.M. and F.S. acknowledge support from Progetti di Ricerca 2017 'Sapienza' Università di Roma (N. RM11715C639F6A69).

## Author contributions

S.C. and C.D.M. designed the study, planned, ran, and analyzed the simulations. S.C. performed the comparison with experiments. C.D.M. developed the models, wrote the codes, and provided numerical tools. All authors contributed to the interpretation of the data. S.C. wrote the manuscript with contributions from C.D.M. and F.S.

## Additional information

**Competing interests:** The authors declare no competing interests.

