## [Peer Review File · Nature Communications]

Reviewer #1 (Remarks to the Author):

The overall theme of the paper is the study of autocatalytic reactions, where the product C, formed by bonding A and B, further catalyzes the A-B-reaction, thus providing a positive feedback mechanism. Such a topic is very intriguing and relevant for modern complex catalytic systems and should appeal to a broad audience. The comparisons to experiments look convincing, potentially providing new mechanistic pictures and interpretations of those. However, I have issues with some of the techniques and parts of the presentation of the work which render the paper confusing and little transparent, I have to say.

I do not understand what patchy particles have to do with all this. In the reaction-diffusion model presented at the beginning and in the supp. Information only the number of reactive bonds come in. Of course, for sterical reasons a realization in the simulations with particles needs directed and anisotropic bonding but I would not call that patchiness in the sense how it is usually used in the field of self-assembly of colloids. The implementation into a simulations model could be done in various ways, maybe even by a lattice model with directed bonding, so why is it called patchiness? In that sense, I find some parts of the paper confusing and they require rewriting.

The clustering behavior falls from the sky at some point and its existence and importance becomes a bit clearer only by cross-reading and jumping around in the manuscript. This behavior should be explained earlier, when the reaction-diffusion model is introduced. Shouldn't it also impact the model?

In the reaction-diffusion model and simulations it seems no back-reaction are allowed, i.e., breaking of bonds. Why?

The wording "diffusion coefficient of the system" is confusing. There should be individual diffusion constants for single molecules and for the clusters, depending on their size. So, the system diffusion is the mean? But is this a proper definition for the use in k_{diff} ? Why?

What are the boundary conditions of the reaction-diffusion model? Is there always an infinitely large amount of reactants? In experimental reality, as the authors say, reactants are consumed and depleted at the end of the reaction. Where is this found in the model in the SI? I guess some steady-state solution is assumed and then applied in the non-equilibrium mean-field or simulation treatment?

That the chemical reaction rate depends globally on p , the total number of formed bonds, may be a sufficient assumption for a mean-field reaction-diffusion theory, but why is this also used in the simulations? In the latter there is access to local geometry and k_{chem} could be used according the local structure, as it should be in experimental reality. A chemical reaction rate is typically always a local quantity (involving an electronic transition, bond forming and breaking, etc.)

Why is the Smoluchowski problem in Suppl Note 2 solved for 1D? These diffusion problems depend qualitatively on dimensionality and it's unclear if the result is transferable from 1D to 3D. Also the boundary conditions far away from the adsorbing sink should be better explained. Also here results depend a lot on boundary conditions, e.g. constant density at infinity, or time-dependent density at a fixed boundary?

Simulation snapshots as illustrations, e.g., for particle and cluster geometries, are in urgent need

Minor:

- reference to a Fig. in the SI is missing on page 2

- absorbing  adsorbing

Reviewer #2 (Remarks to the Author):

Referee report on the manuscript NCOMMS-17-31600: "Exploiting patchy particles to understand autocatalytic kinetics" by Silvia Corezzi, Francesco Sciortino and Cristiano De Michele.

The manuscript addresses a model of patchy particles with autocatalytic ability. The model put forward is – as argued by the authors – a very good model for exploring the physical principles of bio-inspired autocatalytic reactive mixtures; as it includes anisotropy, variable barriers and adjustable autocatalytic ability. The model is kept simple enough so that its aggregation behaviour is amenable to analytical theory and simulations, yet it is also complex enough so that it can be applied to real systems.

They find that the rate of aggregation is appreciably higher than expected (exponential slow down following Boltzmann distribution) due to reduction of the activation barrier (different contributions are discussed in detail and convincingly). This is a novel effect and was not discussed in the literature previously. The comparison with experimental data on polymerization rates of epoxy-amine systems shows excellent agreement and in fact resolves the long-standing open problem.

The topic is clearly important for a vast range of problems in soft matter, chemistry, biology and material science. The nice aspect of it is the universality of the physical mechanisms it describes. It can be easily applied to many specific systems; one example is the polymerization experiments focused on in this work. The manuscript is very well written and in my opinion it should be published as it is in Nature Communications.

Reference: NCOMMS-17-31600

Manuscript title: Exploiting limited valence "patchy" particles to understand autocatalytic kinetics

Response to the comments of reviewer #1

Hereafter, the Referee's comments are in italics, the authors' reply as normal text. Major changes are colored blue in the revised manuscript and in the following.

The overall theme of the paper is the study of autocatalytic reactions, where the product C, formed by bonding A and B, further catalyzes the A-B-reaction, thus providing a positive feedback mechanism. Such a topic is very intriguing and relevant for modern complex catalytic systems and should appeal to a broad audience. The comparisons to experiments look convincing, potentially providing new mechanistic pictures and interpretations of those. However, I have issues with some of the techniques and parts of the presentation of the work which render the paper confusing and little transparent, I have to say.

Authors: We thank the Referee for finding the topic intriguing and our work of interest. More importantly, we thank her/him for providing important suggestions. Our point-by-point response to the raised issues is provided below. We do believe – and this is no empty phrase – that her/his comments have significantly contributed to improve the quality of the manuscript, by forcing us to fill gaps and to make the general presentation clearer.

- 1. I do not understand what patchy particles have to do with all this. In the reaction-diffusion model presented at the beginning and in the supp. information only the number of reactive bonds come in. Of course, for sterical reasons a realization in the simulations with particles needs directed and anisotropic bonding but I would not call that patchiness in the sense how it is usually used in the field of self-assembly of colloids. The implementation into a simulations model could be done in various ways, maybe even by a lattice model with directed bonding, so why is it called patchiness? In that sense, I find some parts of the paper confusing and they require rewriting.*

Authors: This Referee's comment helped us to understand that we were not sufficiently clear in presenting the reasons for using a "patchy" particle model. Indeed, the critical ingredient, as the Referee properly points out is the limited "valence", i.e. a precise control on the number of reactive sites of each particle. Selecting a model in which each patch (due to geometric constraints) can not be involved in more than one bond provides a simple way of implementing the prescribed "valence" and numerically study the aggregation process.

To remedy this situation we have especially worked on several parts of the manuscript. In the introduction we have added:

"Patchy particle models, which are designed learning from nature, can be thought of as an archetype of real monomers and their individual motion is an integral part of the bonding process. While these models have been predominantly used in the field of self-assembly of colloids, their range of applicability is much wider, ranging from associating and network-forming liquids [19,20] to chemically reactive particles [21]. Their flexibility originates from the possibility of

implementing a limited number of reactive bonds by geometrically enforcing the single-bond per patch condition [22].”

We have modified in this regard the manuscript title into “Exploiting limited valence ‘patchy’ particles to understand autocatalytic kinetics”.

Also, on the first part of the Results we have added a new paragraph (page 5), which reads:

“These are the basic autocatalytic scenarios of practical importance to which we refer, and which we aim to reproduce within a simple patchy particle model where bonds irreversibly form between complementary reactive sites and progressively give rise to particle clusters in the form of branched aggregates or networks. It has been previously noticed that owing to the random nature of the growth mechanism in step-wise aggregation, the system's evolution slaved to the bonding process proceeds along a sequence of equilibrium states only controlled by the number of formed bonds [24-27], so that when the fraction of bonds is ρ , the number $n(s)$ of clusters made of s particles can be calculated following statistical mechanics prescriptions [24], independent of the time dependence of ρ . The presence of autocatalytic mechanisms of aggregation, on the other hand, expresses itself in the evolution of ρ over time. Therefore, to find out the rule for a proper design of autocatalytic patchy particles, we first develop a statistical treatment for the rate equation of an autocatalytic, irreversible aggregation process.”

We hope this new paragraph will provide the rationale of what is done and, in particular, explains why we start by developing a statistical treatment (more details are given in the SI) for the rate equation of an autocatalytic, irreversible aggregation process – the reaction-diffusion model, to use the referee’s wording. Modeling such a process (which describes, e.g., epoxy-amine polymerizations) by means of a patchy particle model is our goal. This requires first finding out the *rule* to make particles able to aggregate autocatalytically. To make the presentation more logical, we have moved after this new paragraph the subheading ‘Autocatalytic chemical rate constant’.

Simulating particles with shapes and excluded volumes similar to the real molecules and with small attractive patches mimicking functional sites makes it thus possible to account for the role of the diffusion process on the bonding process and, with minimal complexity, for the functionality of the same molecules. In other words, *patchy* are not the reactive units referred to in the reaction-diffusion model but so are the particles we decided to use to realize the condition derived from the model. In principle, as the Referee noted, the implementation into a simulation model could be done in various ways. However, we think that different implementations would not be equally realistic, as they would miss some aspects related to the ability of particles to explore a continuum of space.

- 2. The clustering behavior falls from the sky at some point and its existence and importance becomes a bit clearer only by cross-reading and jumping around in the manuscript. This behavior should be explained earlier, when the reaction-diffusion model is introduced. Shouldn't it also impact the model?*

Authors: We welcome the Referee's suggestion and now we introduce the existence of clustering behavior in our patchy particle model earlier in the manuscript, when the reaction-diffusion model

is introduced. Indeed, in the new paragraph we have added on page 5, we explain that clustering is inherent in the step-wise mechanism of reaction, and introduce the cluster size distribution $n(s)$ discussed later on in the manuscript:

... and which we aim to reproduce within a simple patchy particle model where bonds irreversibly form between complementary reactive sites. The progressive bonding gives rise to larger and larger particle aggregates in the form of branched clusters or networks.

And

... so that when the fraction of bonds is p , the number $n(s)$ of clusters made of s particles can be calculated following statistical mechanics prescriptions [24], independent of the time dependence of p ."

Furthermore, the following simulation snapshots have been added as Fig.4d of the revised manuscript to illustrate populations of clusters of different size (from monomers to tetramers) grown during the aggregation process.

Fig.4: **System's diffusivity and cluster size distribution.** ... d Populations of clusters of different size, from monomers to tetramers, in the system reacted up to $p=0.5$.

While randomness in bonding pairs of complementary reactive sites is an essential part of the reaction-diffusion model, it should be noted that no information strictly related to clusters is needed within a mean-field approach to derive the rate equation of the aggregation process, but only the assumption of equal *a priori* reactivity of all sites of the same type is required.

3. *In the reaction-diffusion model and simulations it seems no back-reaction are allowed, i.e., breaking of bonds. Why?*

Authors: Our main aim is to model the physical principles of autocatalytic aggregation of reactive mixtures, as exemplified by the process of epoxy-amine cure. The energy of these covalent bonds is significantly larger than the thermal energy $k_B T$ (e.g., the average bonding energy for N–C bonds is ~ 300 kJ/mol, about 120 times the value of thermal energy at ambient T) effectively rendering negligible the back-reaction at ambient temperature. Thus, the reaction equilibrium strongly favors product formation. Indeed in all the experiments we compare to the extent of

reaction at the end exceeds 0.8. Note that ρ does not reach one since the missing bonds cannot form due to the vanishing of the particle diffusivity associated to the formation of a highly connected bond network.

Limiting ourselves to the relevant cases in which the reaction equilibrium strongly favors product formation, we have chosen to study a model that is kept simple enough so that its aggregation behaviour is amenable to simple analytical theory and simulations, yet it is also complex enough so that it can be applied to real systems. We now stress on page 3 of the revised manuscript that we focus on a simple patchy particle model "with a view to comparing the simulation results with experiments of formation of effectively irreversible strong covalent bonds between epoxy and amino reagents".

We like to note here that any simulation model of irreversible aggregation can be transformed into a corresponding reversible model, by simply changing the ratio between bonding energy of reactive sites, u_0 , and the thermal energy, $k_B T$. Therefore, in order to study a model where back-reactions are allowed it is sufficient in the simulations to reduce u_0 and in the reaction-diffusion model to readapt the rate equation with the guidance provided in the new references 25-27. In the revised version we have added a sentence in the Discussion to stress that our model "offers the possibility to implement equally well a reversible (physical) aggregation process, through the modulation of bonding energy of reactive sites, \$u_0\$: once a bond is formed there is a probability of bond breaking which depends on the relative value of the thermal energy compared to the energy barrier that the reacted sites must overcome to escape the potential well [26,27]."

4. *The wording "diffusion coefficient of the system" is confusing. There should be individual diffusion constants for single molecules and for the clusters, depending on their size. So, the system diffusion is the mean? But is this a proper definition for the use in k_{diff} ? Why?*

Authors: We thank the Referee for giving us the possibility to clarify what *diffusion coefficient of the system* is. As described in the Methods, we calculate the diffusion coefficient of a system composed of N particles as the average diffusion coefficient over all the individual particles composing the system. To avoid confusion, we now use the wording "average diffusion coefficient of the particle" in place of "diffusion coefficient of the system" throughout the revised manuscript. Concerning the proper definition of D to be used in k_{diff} , we stress that $1/k_{diff}$ should represent an average time for bonding pairs of reactive sites that is related to diffusion, i.e., to the ability of reactive sites to move through the sample. Although it is an assumption, we find it a reasonable mean-field assumption that $1/k_{diff}$ is proportional to $1/D$ which is itself a characteristic time related to the average diffusion of the particles bearing reactive sites. The appropriateness of the proposed assumption is further supported by the following consideration:

$\tau_{diff} \sim 1/D = (\sum_i D_i/N)^{-1} \Rightarrow k_{diff} = \sum_i k_{i,diff}/N$ with the sum running over all particles in the system. Therefore, the overall diffusive rate constant is the mean of the individual rate constants for sites of all the particles in the system, and this is reasonable because more mobile particles are expected to contribute more to the rate of bonding. Differently, if one assumes for example $\tau_{diff} = \sum_i \tau_{i,diff}/N \sim \sum_i D_i^{-1}/N$, then, the overall diffusive rate constant comes to be controlled by the less mobile particles also in the presence of very mobile ones, which is unreasonable. In conclusion, we think that the average diffusion of the system is the proper definition of D for use

in k_{diff} within our description. Besides that, we also note that a very detailed analysis of the diffusion properties of the model without barriers (see new ref.28) shows that most of the average diffusion coefficient of the system comes from the contribution of monomers which are orders of magnitude more mobile than other cluster sizes and always greater in number.

5. *What are the boundary conditions of the reaction-diffusion model? Is there always an infinitely large amount of reactants? In experimental reality, as the authors say, reactants are consumed and depleted at the end of the reaction. Where is this found in the model in the SI? I guess some steady-state solution is assumed and then applied in the non-equilibrium mean-field or simulation treatment?*

Authors: The reaction-diffusion model is developed for a stoichiometric mixture consisting of N_A monomers with functionality f_A and N_B monomers with functionality f_B within a box of volume V , that is exactly the same mixture that we study in the simulations. The description can be found at the bottom of page 1 in the SI, at the point where we write the probability P_a of finding the autoproduced catalyst within an active distance.

Notice that an erroneous infinite limit in the sum symbol in one equation of the SI (fifth equation from the top) has been corrected.

6. *That the chemical reaction rate depends globally on p , the total number of formed bonds, may be a sufficient assumption for a mean-field reaction-diffusion theory, but why is this also used in the simulations? In the latter there is access to local geometry and k_{chem} could be used according the local structure, as it should be in experimental reality. A chemical reaction rate is typically always a local quantity (involving an electronic transition, bond forming and breaking, etc.)*

Authors: The Referee is correct in stating that a chemical reaction rate is always a local quantity. A full description of chemical reactions involving electronic transitions, in principle amenable with quantum mechanic calculations, would definitively be highly desirable. Still, with present day numerical resources, computational studies of the time dependence of the extent of reaction for systems with a reasonable number of molecules are not feasible.

When we started this project we interrogated ourselves how to model (avoiding quantum mechanics) a local structure-dependent reaction rate. All solutions we came up required a significant number of additional *ad-hoc* parameters (for example the concentration of "ghost" particles catalyzing the chemical reaction, their diffusion coefficient, the probability of generating additional "ghost" particles after a successful bond formation process). We decided that it was first necessary to search for a more transparent modeling of autocatalytic reactions. We settled for the simplest way to realize a particle model that includes the key features of an autocatalytic reactive mixture and which allows to explore the physical mechanisms behind an autocatalytic aggregation process under realistic conditions.

It should be noted that the non-autocatalytic version of the model have proved to be a very good realization of mean-field theory predictions, such as those of Flory-Stockmayer (see new ref. 24), suggesting that its behavior can be successfully described beyond local structure.

7. *Why is the Smoluchowski problem in Suppl Note 2 solved for 1D? These diffusion problems depend qualitatively on dimensionality and it's unclear if the result is transferable from 1D to 3D. Also the boundary conditions far away from the adsorbing sink should be better explained. Also here results depend a lot on boundary conditions, e.g. constant density at infinity, or time-dependent density at a fixed boundary?*

Authors: We have to apologize for not being clear in the presentation of the Smoluchowski problem solved in Supplementary Note 2. Our treatment to study diffusion-limited irreversible cluster aggregation builds on the modeling proposed in F. Sciortino, A. Belloni, and P. Tartaglia, Phys. Rev. E 52, 4068 (1995). Within such approach, one cluster is assumed to be at the origin and the time evolution of the number density of clusters is studied via the 3D Smoluchowski equation coupled to an equation that controls the mass growth of clusters. Since the cluster at the origin is spherical, the Smoluchowski equation can be conveniently written in spherical coordinates with the number density depending only on the radial distance from the origin r . Therefore, the original 3D Smoluchowski equation is recasted into a 1D differential equation for the r -dependent number density. With respect to the original work of Sciortino *et al.* we also consider an entry barrier for the cluster aggregation (which is modeled as an isotropic external potential), mimicking the entry barrier present in the site-site interaction between patchy particles used in the molecular dynamics simulations (see Fig. 1 in the manuscript and the corresponding discussion in the text).

We have realized that we generated some confusion in the text by saying (few lines after Eq. 2) that *"we solved the Smoluchowski equation of diffusion in a simplified case, i.e. in one dimension for particles in the presence of a growing sink (as if due to clustering) with a barrier for absorption (Supplementary Note 2)"*, and also the notation x for the distance between clusters, we admit it, was a poor choice. Hence, we apologize for this, and in the revised manuscript we have replaced the above text with the following sentence:

"we consider a model, based on the solution of Smoluchowski equation of diffusion in three dimensions [29], which aims to describe in a simplified way the growth process of a sink (due to clustering) in the presence of a barrier for particle adsorption (see Supplementary Note 2)."

In addition, we have made substantial changes to the Supplementary Note 2 to describe more clearly what we did and to avoid any confusion.

Concerning the boundary conditions which we employed to solve the Smoluchowski equation, as in the study of F. Sciortino *et al.* Phys. Rev. E 52, 4068 (1995), we used *adsorption boundary conditions* on the surface of the spherical cluster at the origin which acts as an adsorbing sink (since we are modeling irreversible cluster aggregation), and *constant density* at infinity as now pointed out in the new version of Supplementary Note 2.

We finally note that we made use of this very simplified model based on the Smoluchowski equation just to highlight the fact that a *barrier effect* can already emerge in a minimalist modeling

of cluster aggregation. For this reason, it was not unexpected to find a clear signature of it in the more realistic autocatalytic patchy particle model that we studied in the present work.

8. *Simulation snapshots as illustrations, e.g., for particle and cluster geometries, are in urgent need*

Authors: Following the Referee's request, we have added in the revised manuscript two simulation snapshots as illustrations, for particles (new Fig. 1, see below) and for cluster populations (new Fig. 4d, previously shown) – we apologize for this lack of illustrative clarity in the previous version of the manuscript.

Fig.1: **The model.** Graphic description of the two types of hard ellipsoids composing the simulated system, and snapshot of the system in the initially monomeric state. The centers of the small (yellow) spheres locate the bonding sites on the surface of the hard-core particle. The site-site interaction, sketched as a function of the inter-site distance r , is modeled as an attractive square-well potential complemented by a repulsive entry barrier of finite height (in red) and an infinite repulsive exit barrier (in green). Notice that bonds can be formed only between small (blue) and big (cyan) particles

Minor:

- reference to a Fig. in the SI is missing on page 2
- absorbing  adsorbing

Authors: We thank the Referee for noting these minor errors that have been corrected in the revised version of the manuscript.

Reviewer #2 (Remarks to the Author):

Referee report on the manuscript NCOMMS-17-31600: "Exploiting patchy particles to understand autocatalytic kinetics" by Silvia Corezzi, Francesco Sciortino and Cristiano De Michele.

The manuscript addresses a model of patchy particles with autocatalytic ability. The model put forward is – as argued by the authors – a very good model for exploring the physical principles of bio-inspired autocatalytic reactive mixtures; as it includes anisotropy, variable barriers and adjustable autocatalytic ability. The model is kept simple enough so that its aggregation behaviour is amenable to analytical theory and simulations, yet it is also complex enough so that it can be applied to real systems.

They find that the rate of aggregation is appreciably higher than expected (exponential slow down following Boltzmann distribution) due to reduction of the activation barrier (different contributions are discussed in detail and convincingly). This is a novel effect and was not discussed in the literature previously. The comparison with experimental data on polymerization rates of epoxy-amine systems shows excellent agreement and in fact resolves the long-standing open problem.

The topic is clearly important for a vast range of problems in soft matter, chemistry, biology and material science. The nice aspect of it is the universality of the physical mechanisms it describes. It can be easily applied to many specific systems; one example is the polymerization experiments focused on in this work. The manuscript is very well written and in my opinion it should be published as it is in Nature Communications.

Authors: We are grateful to the Referee for appreciating our work and supporting publication in *Nature Communications*.

Reviewer #1 (Remarks to the Author):

The authors have addressed satisfactorily my questions and the paper has improved much in clarity and presentation. It reports novel mechanisms important for a wide field in natural and material science and I recommend publication now as it is.

Reference: NCOMMS-17-31600A

Manuscript title: Exploiting limited valence 'patchy' particles to understand autocatalytic kinetics

Response to the comments of reviewer #1

The authors have addressed satisfactorily my questions and the paper has improved much in clarity and presentation. It reports novel mechanisms important for a wide field in natural and material science and I recommend publication now as it is.

Authors: We are grateful to the Referee who provided valuable suggestions for significantly improving the paper and now supports its publication in *Nature Communications*.